# Base-Promoted Intramolecular Addition of Vinyl Cyclopropanecarboxamides to Access Conformationally Restricted Aza[3.1.0]bicycles

**DOI:** 10.3390/molecules28093691

**Published:** 2023-04-25

**Authors:** Jingya Li, Zhiguo Zhang, Liming Chen, Mengjuan Li, Xingjie Zhang, Guisheng Zhang

**Affiliations:** Key Laboratory of Green Chemical Media and Reactions, Ministry of Education, Henan Key Laboratory of Organic Functional Molecules and Drug Innovation, Collaborative Innovation Center of Henan Province for Green Manufacturing of Fine Chemicals, School of Chemistry and Chemical Engineering, Henan Normal University, Xinxiang 453007, China

**Keywords:** aza[3.1.0]bicycles, vinyl cyclopropanecarboxamides, amination, addition of alkenes

## Abstract

3-Azabicyclo[3.1.0]hexanes are common structural components in natural products and bioactive compounds. Traditionally, the metal-mediated cyclopropanation domino reaction of chain enzymes is the most commonly used strategy for the construction of this type of aza[3.1.0]bicycle derivative. In this study, a base-promoted intramolecular addition of alkenes used to deliver conformationally restricted highly substituted aza[3.1.0]bicycles is reported. This reaction was tailor-made for saturated aza[3.1.0] bicycle-containing fused bicyclic compounds that may be applied in the development of concise and divergent total syntheses of bioactive compounds.

## 1. Introduction

Saturated *N*-heterocycles such as 3-azabicyclo[3.1.0]hexanes are common structural components in natural products and bioactive compounds with a broad spectrum of activity against various bacteria, mycobacteria, parasites, tumors, and neurological disorders (Figure 1) [1,2,3,4,5,6,7,8,9]. For example, Duocarmycin SA, Yatakemycin, and CC-1065 are representative members of such well-known biomolecules that derive their antitumor activity from their ability to alkylate DNA [10]. Furthermore, they have been identified as useful synthons in a range of organic transformations [11,12,13,14,15,16,17,18,19]. Consequently, the development of methods enabling the efficient construction of such structures has been a research focus in organic chemistry.

A literature review indicates that the metal-mediated cyclopropanation domino reaction of chain enynes is the most commonly used strategy for the construction of aza[3.1.0]bicycle derivatives in terms of scalability and substrate scope, which highly rely on the in situ-generated metal carbene species in the presence of Pd, Au, Ru, Co, Ni, and Rh salts as catalysts [20,21,22,23,24,25,26,27,28,29,30,31,32,33]. Occasionally, the same conversion starting from enyne analogues has also been achieved by a photocatalytic pathway [34,35,36] as well as metal-free organocatalytic processes [37,38,39,40,41,42], mechanisms that are similar to the metal carbene processes (Figure 1a). Another two effective approaches for the synthesis of 3-azabicyclo[3.1.0]hexanes involve the derivatization reactions of substituted cyclopropanes, such as C(sp^3^)–H bond activated alkenylation/amination tandem reactions and intramolecular aminolysis reactions (Figure 1b) [43,44,45,46,47,48,49,50,51,52,53], and the reaction of functionalized maleimide derivatives with one carbon donor generated in situ derived from substituted diazomethanes, bromo(nitro)methane, substituted α-diazoacetates, and *N*-tosylhydrazones via an intermolecular [2+1] fused-annulation reaction (Figure 1c) [54,55,56,57,58,59,60,61,62,63,64]. In particular, the base-induced intramolecular spirocyclization method of the alkylation subunit precursor appeared to be a more efficient proprietary reaction to access 3-azabicyclo[3.1.0]hexane scaffold-containing natural products via aryl metal or radical dearomatization/cyclization reactions (Figure 1d) [65,66,67,68,69,70,71,72,73,74,75,76,77]. Although remarkable processing has been achieved in the last decades, achieving the synthesis of the structurally versatile aza[3.1.0]bicycles through readily available starting materials and simple and efficient chemical transformation remains a challenge. Here, describe our recent effort on the base-promoted intramolecular addition of vinyl cyclopropanecarboxamides **1** to access conformationally restricted aza[3.1.0]bicycles core **2** (Figure 1e).

## 2. Results and Discussion

### 2.1. Reaction Optimization

Very recently, we developed a palladium(II)-catalyzed intramolecular oxidative aza-Wacker-type reaction to access a series of highly substituted aza[3.1.0]bicycles, starting from readily available compounds **1**. Combined with our other works related to the derivatization reactions of amides and previous reports, we envisioned that compound **1** may continue to generate highly substituted aza[3.1.0] bicycles **2** via a molecular olefin aza-addition reaction under appropriate bases (Figure 1e). With this assumption in mind, the model reagent 1-(4-chlorophenyl)-*N*-(*p*-tolyl)-2-vinyl cyclopropane-1-carboxamide (**1a**) was selected to explore the feasibility of the designed transformation; some key results are listed in Table 1. After many attempts, we found that the desired product 1-(4-chlorophenyl)-4-methyl-3-(*p*-tolyl)-3-azabicyclo[3.1.0]hexan-2-one (**2a**) was isolated in 82% yield in the presence of 4.0 equiv. of *^t^*BuOK in DMF after 24 h, along with 11% of recovered **1a**, which could not be consumed by prolonging the reaction time (Table 1, entry 1). Notably, when we added 4.2 equiv. of 18-crown-6 ether to the reaction [48,78], starting material **1a** was completely consumed within 24 h (Table 1, entry 2). However, considering that it did not significantly affect the reaction time and the yield of product **2a**, as well as the economy of the transformation, it was not added in the later experiments. Moreover, reactions performed at a lower or higher loading of *^t^*BuOK failed to give a higher yield of fuse-heterocycle **2a** (Table 1, entries 3–5). Similarly, lower or higher temperatures did not help improve the reaction efficiency (Table 1, entries 6–9). The yield of the target product **2a** was not increased when the reactions were carried out in the presence of four other types of bases, namely, K_3_PO_4_, NaH, NaOH, and Cs_2_CO_3_ (Table 1, entries 10–13). Other solvents, including MeCN, dioxane, toluene, NMP, and DMSO, all provided diminished or no yields of the product (Table 1, entries 14–18).

### 2.2. Substrate Scope

With the identified optimal reaction conditions in hand, we evaluated the scope and drawbacks of this base-promoted intramolecular addition (Figure 2). The variation in R^1^ was examined first. A variety of aryl groups having electron-releasing, -neutral, or -withdrawing groups at the 3- or 4-position of the benzene ring underwent smooth intramolecular annulation leading to the formation of the aza[3.1.0]bicycles **2a**–**g** in 40–85% yields with the regioselectivity ratio ranging from 1:1 to 2:1. Unfortunately, the analogous α-naphthyl-based substrate **1h** was not suitable for this system. *N*-Alkyl-substituted starting material **1i** afforded the desired product **2i** with excellent yield (85%) in *ca* 5:4 of dr value.

Next, the scope of the reaction was evaluated using different R^2^. Selected examples are presented in Figure 3. It can be seen that the addition reaction was proved to be well tolerated by various 1-aryl-substituted vinyl cyclopropanecarboxamides bearing a MeO– (**2j** and **2k**), Me– (**2l**–**n**), F– (**2p**), and Br– (**2q**) group at the *para*-, *meta*-, or *ortho*-position, along with the phenyl group-substituted cyclopropane derivative (**2o**). Notably, the bromobenzene moiety of product **2q** retains a derivatization site for further functionalization reactions, including Suzuki–Miyaura [15,79,80,81,82], Buchwald–Hartwig [83,84,85], and Sonogashi coupling reactions [86,87,88,89,90]. In particular, the starting material **1r** with a styrene group on the cyclopropyl moiety provided the product **2r** with an 81% yield.

With the aim of devising a practical, gram-scale synthesis of a biovaluable aza[3.1.0]bicycle scaffold, a reaction on 7 mmol (1.841 g) was carried out with this improved synthetic method based on the base-promoted intramolecular addition of alkenes. When we treated **1o** under optimal conditions, the reaction smoothly furnished a 73% yield of **2o** after 72 h under standard conditions, with 17% **1o** recovered (86% yield of **2o** based on the conversion of the substrate) (Figure 4).

## 3. Materials and Methods

### 3.1. General Remarks

Unless stated otherwise, reactions were conducted in Schlenk under air. All reagents were purchased from commercial sources and used without further treatment unless otherwise indicated. Starting materials were synthesized following the literatures [91,92,93], and the procedures were described in the Appendix A. DMF, CH_3_CN, DMSO, THF, and toluene for reactions were distilled under an atmosphere of dry N_2_. Petroleum ether (PE), used here, refers to the 60–90 °C boiling point fraction of petroleum. Ethyl acetate is abbreviated as EA. ^1^H NMR and ^13^C NMR spectra were recorded on a Bruker Avance/600 (^1^H: 600 MHz, ^13^C: 151 MHz) or Bruker Avance/400 (^1^H: 400 MHz, ^13^C: 101 MHz at 25 °C). Fluorine nuclear magnetic resonance (^19^F NMR) spectra were recorded on a Bruker Avance/600 spectrometer or a Bruker Avance/400. ^1^H NMR spectra were calibrated against residual CHCl_3_ in the solvent (7.26 ppm). ^13^C NMR spectra were calibrated against the peak of the residual CHCl_3_ in the solvent (77.2 ppm). NMR data are represented as follows: chemical shift (ppm), multiplicity (s = singlet, d = doublet, t = triplet, q = quartet, and m = multiplet), coupling constant in hertz (Hz), and integration. All high-resolution mass spectra (HRMS) were measured on a mass spectrometer by using electrospray ionization orthogonal acceleration time-of-flight (ESI-OA-TOF), and the purity of all samples used for HRMS (>95%) was confirmed by ^1^H NMR and ^13^C NMR spectroscopic analysis. All reactions were monitored by thin-layer chromatography (TLC) (PE: EA = 10:1) with GF254 silica gel-coated plates.

### 3.2. Typical Experimental Procedure for ***2*** (***2a*** as an Example)

In a Schlenk tube (25 mL), **1a** (156 mg, 0.5 mmol) and *^t^*BuOK (224 mg, 4.0 equiv.) were added. The mixture was stirred well in DMF (2 mL) and stirred at 110 °C in a sand bath under air (the whole process was closely monitored by TLC). After the completion of the reaction, DCM (5 mL) was added to water (10 mL) and extracted with dichloromethane (3 × 10 mL). Then the organic solvent was washed with H_2_O (15 mL) and saturated NaCl (15 mL) solutions, dried over anhydrous Na_2_SO_4,_ and concentrated in vacuo. The residue was purified by flash column chromatography with PE and EA (PE: EA = 20: 1) as eluent to give **2a** as a white solid (128 mg, 82%).

### 3.3. Characterization of Products

1-(4-Chlorophenyl)-4-methyl-3-(*p*-tolyl)-3-azabicyclo[3.1.0]hexan-2-one (**2a**). White solid. (Yield: 82%). Mp = 74–76 °C. dr ≈ 1:1. ^1^H NMR (600 MHz, CDCl_3_) *δ* 7.44–7.40 (m, 4H, Ar-H), 7.34–7.30 (m, 4H, Ar-H), 7.29 (dt, *J* = 9.0, 2.4 Hz, 2H, Ar-H), 7.17 (t, *J* = 7.2 Hz, 4H, Ar-H), 7.12 (d, *J* = 8.4 Hz, 2H, Ar-H), 4.53 (p, *J* = 6.0 Hz, 1H, N-CH), 4.20 (q, *J* = 6.4 Hz, 1H, N-CH), 2.38–2.35 (m, 2H, CH), 2.33 (s, 3H, Ar-CH_3_), 2.33 (s, 3H, Ar-CH_3_), 2.04 (dd, *J* = 7.8, 4.8 Hz, 1H, CH), 1.51 (dd, *J* = 7.8, 4.8 Hz, 1H, CH_2_), 1.39 (dd, *J* = 7.8, 4.8 Hz, 1H, CH_2_), 1.36 (d, *J* = 6.0 Hz, 3H, CH_3_), 1.31 (t, *J* = 4.5 Hz, 1H, CH_2_), 1.26 (t, *J* = 4.5 Hz, 1H, CH_2_), 1.19 (d, *J* = 6.0 Hz, 3H, CH_3_). ^13^C NMR (101 MHz, CDCl_3_) *δ* 173.5 (C=O), 172.4 (C=O), 135.8, 135.3, 135.1, 135.0, 134.9, 134.1, 133.1, 132.8, 130.1, 129.8, 129.61, 129.58, 128.6, 128.5, 124.6, 123.5, 56.4 (C-N), 53.1 (C-N), 34.4 (C), 33.4 (C), 26.9 (CH), 26.6 (CH), 21.5 (Ar-CH_3_), 21.0 (Ar-CH_3_), 20.9 (CH_3_), 20.1 (CH_3_), 16.9 (CH_2_), 16.8 (CH_2_). HRMS (ESI) (*m/z*) calculated for C_19_H_18_ClNO [M + Na]^+^: 334.0969, found: 334.0968. IR *v*/cm^−1^ (KBr) 1678, 1512, 1496, 1391, 1396, 1292, 1182, 1086, 1012, 839, 756, 718, 521.3-(4-(*Tert*-butyl)phenyl)-1-(4-chlorophenyl)-4-methyl-3-azabicyclo[3.1.0]hexan-2-one (**2b**). White solid. (Yield: 81%). Mp = 119–121 °C. dr ≈ 5:3. ^1^H NMR (600 MHz, CDCl_3_) δ 7.42 (dd, *J* = 8.4, 3.6 Hz, 3.3H, Ar-H), 7.39 (s, 0.6H, Ar-H), 7.37 (s, 4.3H, Ar-H), 7.33 (d, *J* = 8.4 Hz, 2H, Ar-H), 7.29 (d, *J* = 8.4 Hz, 1.2H, Ar-H), 7.15 (d, *J* = 8.4 Hz, 1.2H, Ar-H), 4.55 (p, *J* = 6.0 Hz, 0.6H, minor, N-CH), 4.23 (q, *J* = 6.0 Hz, 1H, major, N-CH), 2.36 (dt, *J* = 7.8, 4.8 Hz, 0.6H, minor, CH), 2.04 (dd, *J* = 7.8, 4.2 Hz, 1H, major, CH), 1.51 (dd, *J* = 7.8, 4.8 Hz, 1H, major, CH_2_), 1.41 (dd, *J* = 7.8, 4.8 Hz, 0.8H, minor, CH_2_), 1.38 (d, *J* = 6.6 Hz, 3H, major, CH_3_), 1.32 (d, *J* = 4.8 Hz, 1H, minor, CH_2_), 1.31 (s, 14H, (CH_3_)_3_), 1.25 (t, *J* = 4.5 Hz, 1.3H, major, CH_2_), 1.21 (d, *J* = 6.0 Hz, 2H, minor, CH_3_). ^13^C NMR (151 MHz, CDCl_3_) δ 173.5 (minor, C=O), 172.4 (major, C=O), 148.8 (minor), 148.3 (major), 135.1 (major), 135.01 (major), 134.98 (minor), 134.0 (minor), 133.1 (major), 132.8 (minor), 130.1 (major), 129.7 (minor), 128.6 (major), 128.5 (minor), 125.9 (major), 125.8 (minor), 124.1 (minor), 122.9 (major), 56.2 (major, C-N), 53.0 (minor, C-N), 34.51 (minor, Ar-(CH_3_)_3_), 34.46 (major, Ar-(CH_3_)_3_), 34.4 (major, C), 33.4 (minor, C), 31.3 ((CH_3_)_3_), 27.1 (minor, CH), 26.5 (major, CH), 21.6 (major, CH_3_), 20.1 (minor, CH_3_), 17.0 (major, CH_2_), 16.8 (minor, CH_2_). HRMS (ESI) (*m/z*) calculated for C_22_H_24_ClNO [M + Na]^+^: 376.1439, found: 376.1431. IR *v*/cm^−1^ (KBr) 1675, 1515, 1493, 1374, 1291, 1266, 1188, 1103, 1067, 1011, 834, 796, 728, 550.1-(4-Chlorophenyl)-3-(4-methoxyphenyl)-4-methyl-3-azabicyclo[3.1.0]hexan-2-one (**2c**). White solid. (Yield: 85%). Mp = 98–100 °C. dr ≈ 1:1. ^1^H NMR (400 MHz, CDCl_3_) *δ* 7.45–7.40 (m, 4H, Ar-H), 7.35–7.27 (m, 6H, Ar-H), 7.15–7.10 (m, 2H, Ar-H), 6.93–6.87 (m, 4H, Ar-H), 4.48 (p, *J* = 6.0 Hz, 1H, N-CH), 4.12 (q, *J* = 6.0 Hz, 1H, N-CH), 3.80 (s, 6H, Ar-OCH_3_), 2.39–2.32 (m, 1H, CH), 2.04 (dd, *J* = 7.6, 4.4 Hz, 1H, CH), 1.51 (dd, *J* = 7.6, 4.4 Hz, 1H, CH_2_), 1.38 (dd, *J* = 7.8, 5.0 Hz, 1H, CH_2_), 1.34 (d, *J* = 6.0 Hz, 3H, CH_3_), 1.30 (t, *J* = 4.8 Hz, 1H, CH_2_), 1.27 (t, *J* = 4.6 Hz, 1H, CH_2_), 1.17 (d, *J* = 6.4 Hz, 3H, CH_3_). ^13^C NMR (101 MHz, CDCl_3_) *δ* 173.5 (C=O), 172.4 (C=O), 157.7, 157.5, 135.04, 134.97, 133.1, 132.8, 130.4, 130.1, 129.7, 129.6, 128.6, 128.5, 126.4, 125.6, 114.3, 114.3, 56.8 (Ar-OCH_3_), 55.5 (C-N), 53.5 (Ar-OCH_3_), 34.2 (C), 33.4 (C), 26.8 (CH), 26.7 (CH), 21.5 (CH_3_), 20.2 (CH_3_), 17.0 (CH_2_), 16.9 (CH_2_). HRMS (ESI) (*m/z*) calculated for C_19_H_18_ClNO_2_ [M + Na]^+^: 350.0918, found: 350.0912. IR *v*/cm^−1^ (KBr) 1674, 1512, 1497, 1375, 1300, 1181, 1110, 1088, 1030, 832, 755, 716, 537.1-(4-Chlorophenyl)-3-(3-methoxyphenyl)-4-methyl-3-azabicyclo[3.1.0]hexan-2-one (**2d**). White solid. (Yield: 51%). Mp = 78–80 °C. dr ≈ 5:3. ^1^H NMR (400 MHz, CDCl_3_) δ 7.44–7.39 (m, 3H, Ar-H), 7.35–7.27 (m, 3.7H, Ar-H), 7.26–7.22 (m, 1.8H, Ar-H), 6.96 (dd, *J* = 8.0, 1.2 Hz, 1H, major, Ar-H), 6.85 (t, *J* = 2.2 Hz, 0.6H, minor, Ar-H), 6.80 (dd, *J* = 8.0, 1.2 Hz, 0.6H, minor, Ar-H), 6.78–6.70 (m, 1.5H, Ar-H), 4.56 (p, *J* = 6.0 Hz, 0.6H, minor, N-CH), 4.27 (q, *J* = 6.4 Hz, 1H, major, N-CH), 3.80 (s, 4.5H, Ar-OCH_3_), 2.38 (dt, *J* = 7.9, 5.0 Hz, 0.6H, minor, CH), 2.06 (dd, *J* = 8.0, 4.4 Hz, 1H, major, CH), 1.51 (dd, *J* = 7.6, 4.8 Hz, 1H, major, CH_2_), 1.44–1.38 (m, 3.7H), 1.32 (t, *J* = 4.8 Hz, 1H, major, CH_2_), 1.26 (t, *J* = 4.4 Hz, 1.8H, CH_2_), 1.23 (d, *J* = 6.4 Hz, 2H, minor, CH_3_). ^13^C NMR (101 MHz, CDCl_3_) δ 173.5 (minor, C=O), 172.6 (major, C=O), 160.13 (major), 160.07 (minor), 139.2 (major), 138.0 (minor), 134.82 (minor), 134.81 (major), 133.2 (major), 132.9 (minor), 130.3 (major), 129.9 (minor), 129.60 (major), 129.55 (minor), 128.6 (major), 128.5 (minor), 116.5 (minor), 114.6 (major), 111.7 (minor), 111.1 (major), 110.6 (minor), 108.8 (major), 56.1 (minor, Ar-OCH_3_), 55.4 (major, C-N), 53.1 (major, Ar-OCH_3_), 34.8 (minor, C), 33.6 (major, C), 26.9 (minor, CH), 26.3 (major, CH), 21.5 (major, CH_3_), 20.0 (minor, CH_3_), 16.9 (major, CH_2_), 16.7 (minor, CH_2_). HRMS (ESI) (*m/z*) calculated for C_19_H_18_ClNO_2_ [M + Na]^+^: 350.0918, found: 350.0914. IR *v*/cm^−1^ (KBr) 1681,1602, 1579, 1490, 1456, 1373, 1293, 1173, 1087, 1068, 1037, 848, 756, 570.1-(4-Chlorophenyl)-4-methyl-3-phenyl-3-azabicyclo[3.1.0]hexan-2-one (**2e**). White solid. (Yield: 64%). Mp = 88–90 °C. dr ≈ 5:3. ^1^H NMR (400 MHz, CDCl_3_) δ 7.49–7.45 (m, 2H, Ar-H), 7.43 (d, *J* = 8.4 Hz, 3H, Ar-H), 7.40–7.37 (m, 1.3H, Ar-H), 7.35 (d, *J* = 9.2 Hz, 2.8H, Ar-H), 7.33–7.28 (m, 2H, Ar-H), 7.25–7.21 (m, 1.3H, Ar-H), 7.18 (t, *J* = 7.4 Hz, 1.3H, Ar-H), 4.59 (p, *J* = 6.0 Hz, 0.6H, minor, N-CH), 4.27 (q, *J* = 6.4 Hz, 1H, major, N-CH), 2.42–2.35 (m, 0.6H, minor, CH), 2.06 (dd, *J* = 7.6, 4.4 Hz, 1H, major, CH), 1.52 (dd, *J* = 7.6, 4.8 Hz, 1H, major, CH_2_), 1.43–1.40 (m, 0.5H, minor, CH_2_), 1.38 (d, *J* = 6.0 Hz, 3H, major, CH_3_), 1.33 (t, *J* = 4.8 Hz, 0.7H, minor, CH_2_), 1.27 (t, *J* = 4.6 Hz, 1.5H, major, CH_2_), 1.21 (d, *J* = 6.0 Hz, 2H, minor, CH_3_). ^13^C NMR (151 MHz, CDCl_3_) δ 173.5 (minor, C=O), 172.5 (major, C=O), 137.8 (major), 136.8 (minor), 134.9 (major), 134.8 (minor), 133.2 (major), 132.9 (minor), 130.1 (major), 129.8 (minor), 129.0 (major), 128.9 (minor), 128.6 (major), 128.5 (minor), 125.9 (minor), 125.4 (minor), 124.5 (major), 123.2 (major), 56.1 (major, C-N), 53.0 (minor, C-N), 34.6 (major, C), 33.5 (minor, C), 26.9 (minor, CH), 26.5 (major, CH), 21.5 (major, CH_3_), 20.1 (minor, CH_3_), 16.9 (major, CH_2_), 16.8 (minor, CH_2_). HRMS (ESI) (*m/z*) calculated for C_18_H_16_ClNO [M + Na]^+^: 320.0813, found: 320.0807. IR *v*/cm^−1^ (KBr) 1680, 1595, 1491, 1374, 1294, 1178, 1102, 1065, 1039, 838, 753, 719, 528.1-(4-Chlorophenyl)-3-(4-fluorophenyl)-4-methyl-3-azabicyclo[3.1.0]hexan-2-one (**2f**). White solid. (Yield: 61%). Mp = 82–84 °C. dr ≈ 3:2. ^1^H NMR (400 MHz, CDCl_3_) δ 7.44–7.36 (m, 5.3H, Ar-H), 7.35–7.31 (m, 2H, Ar-H), 7.31–7.28 (m, 2H, Ar-H), 7.22–7.16 (m, 1.3H, Ar-H), 7.11–7.02 (m, 3.2H, Ar-H), 4.52 (p, *J* = 6.0 Hz, 0.7H, minor, N-CH), 4.19 (q, *J* = 6.4 Hz, 1H, major, N-CH), 2.39 (dt, *J* = 8.0, 4.8 Hz, 0.7H, minor, CH), 2.07 (dd, *J* = 7.6, 4.4 Hz, 1H, major, CH), 1.53 (dd, *J* = 7.6, 4.8 Hz, 1H, major, CH_2_), 1.41 (dd, *J* = 7.6, 4.8 Hz, 1H, major, CH_2_), 1.36 (d, *J* = 6.0 Hz, 3H, major, CH_3_), 1.31 (t, *J* = 4.8 Hz, 0.8H, minor, CH_2_), 1.27 (t, *J* = 4.8 Hz, 2H, CH_2_), 1.20 (d, *J* = 6.4 Hz, 2H, minor, CH_3_). ^13^C NMR (151 MHz, CDCl_3_) δ 173.6 (minor, C=O), 172.5 (major, C=O), 160.5 (d, *J* = 246.1 Hz, minor, C-F), 160.2 (d, *J* = 244.6 Hz, major, C-F), 134.7 (major), 134.6 (minor), 133.6 (d, *J* = 3.0 Hz, major), 133.3 (major), 133.0 (minor), 132.7 (d, *J* = 2.7 Hz, minor), 130.1 (major), 129.8 (minor), 128.6 (major), 128.5 (minor), 126.4 (d, *J* = 4.5 Hz, minor), 125.4 (d, *J* = 7.6Hz, major), 115.9 (d, *J* = 4.5 Hz, minor), 115.8 (d, *J* = 6.0 Hz, major), 56.5 (major, N-CH), 53.3 (minor, N-CH), 34.3 (major, C), 33.4 (minor, C), 26.8 (minor, CH), 26.5 (major, CH), 21.4 (major, CH_3_), 20.1 (minor, CH_3_), 16.9 (major, CH_2_), 16.8 (minor, CH_2_). ^19^F NMR (565 MHz, CDCl_3_) δ −115.8 (minor), −116.4 (major). HRMS (ESI) (*m*/*z*) calculated for C_18_H_15_ClFNO [M + Na]^+^: 338.0718, found: 338.0716. IR *v*/cm^−1^ (KBr) 1682, 1505, 1378, 1180, 1101, 1064, 1014, 833, 718, 533.1-(4-Chlorophenyl)-3-(3-fluorophenyl)-4-methyl-3-azabicyclo[3.1.0]hexan-2-one (**2g**). White solid. (Yield: 40%). Mp = 82–84 °C. dr ≈ 2:1. ^1^H NMR (400 MHz, CDCl_3_) δ 7.43–7.38 (m, 3.8H, Ar-H), 7.36–7.34 (m, 1.5H, Ar-H), 7.33–7.31 (m, 1.7H, Ar-H), 7.30 (d, *J* = 1.6 Hz, 0.5H, Ar-H), 7.29 (d, *J* = 3.6 Hz, 0.5H, Ar-H), 7.27–7.26 (m, 0.6H, Ar-H), 7.25–7.23 (m, 0.4H, Ar-H), 7.03 (dd, *J* = 8.8, 1.2 Hz, 1H, Ar-H), 6.93–6.83 (m, 1.4H, Ar-H), 4.57 (p, *J* = 6.0 Hz, 0.5H, minor, N-CH), 4.28 (q, *J* = 6.4 Hz, 1H, major, N-CH), 2.45–2.38 (m, 0.4H, minor, CH), 2.07 (dd, *J* = 8.0, 4.4 Hz, 1H, major, CH), 1.53 (dd, *J* = 7.6, 4.8 Hz, 1H, major, CH_2_), 1.45–1.39 (m, 3.5H), 1.32 (t, *J* = 4.6 Hz, 0.6H, minor, CH_2_), 1.28–1.22 (m, 3H). ^13^C NMR (151 MHz, CDCl_3_) δ 173.5 (minor, C=O), 172.6 (major, C=O), 163.0 (d, *J* = 246.1 Hz, major, C-F), 162.9 (d, *J* = 246.1 Hz, minor, C-F), 139.6 (d, *J* = 10.6 Hz, major), 138.4 (d, *J* = 10.6 Hz, minor), 134.5 (d, *J* = 3.0 Hz, major), 133.4 (major), 133.1 (minor), 130.2 (minor), 130.1 (d, *J* = 9.1 Hz, major), 130.0 (d, *J* = 9.1 Hz, minor), 129.9 (major), 128.7 (major), 128.5 (minor), 119.6 (d, *J* = 3.0 Hz, minor), 117.4 (d, *J* = 3.0 Hz, major), 112.6 (d, *J* = 6.0 Hz, minor), 111.8 (d, *J* = 21.1 Hz, major), 111.5, 109.8 (d, *J* = 25.7 Hz, major), 55.9 (major, N-CH), 52.9 (minor, N-CH), 34.8 (major, C), 33.6 (minor, C), 26.9 (minor, CH), 26.3 (major, CH), 21.3 (major, CH_3_), 20.0 (minor, CH_3_), 16.9 (major, CH_2_), 16.7 (minor, CH_2_). ^19^F NMR (565 MHz, CDCl_3_) δ -111.3 (major), -111.8 (minor). HRMS (ESI) (*m/z*) calculated for C_18_H_15_ClFNO [M + Na]^+^: 338.0718, found: 338.0712. IR *v*/cm^−1^ (KBr) 1682, 1588, 1491, 1452, 1368, 1296, 1185, 1087, 1064, 1014, 856, 754, 719, 591, 482.3-Butyl-1-(4-chlorophenyl)-4-methyl-3-azabicyclo[3.1.0]hexan-2-one (**2i**). yellow oil. (Yield: 85%) °C. dr ≈ 5:4. ^1^H NMR (600 MHz, CDCl_3_) δ 7.35 (dd, *J* = 10.8, 8.4 Hz, 3H, Ar-H), 7.28 (d, *J* = 7.8 Hz, 3H, Ar-H), 3.98 (p, *J* = 6.0 Hz, 1H, minor, N-CH), 3.61–3.55 (m, 1.5H, minor), 3.54–3.48 (m, 1H, major, N-CH), 2.90–2.83 (m, 1.6H, major), 2.21–2.16 (m, 1H, major, CH), 1.86 (dd, *J* = 7.8, 4.2 Hz, 1H, minor, CH), 1.52–1.45 (m, 1.5H, major), 1.45–1.42 (m, 1H, minor), 1.41 (dd, *J* = 7.8, 4.8 Hz, 1.6H, major), 1.32 (d, *J* = 6.0 Hz, 2.7H, minor, CH_3_), 1.31–1.29 (m, 2H), 1.28–1.26 (m, 1.3H), 1.25 (d, *J* = 6.6 Hz, 3H, major, CH_3_), 1.08 (t, *J* = 4.8 Hz, 1H, major, CH_2_), 0.99 (t, *J* = 4.2 Hz, 0.8H, minor, CH_2_), 0.95–0.90 (m, 5H). ^13^C NMR (151 MHz, CDCl_3_) δ 173.9 (major, C=O), 173.2 (minor, C=O), 135.4 (minor), 135.3 (major), 132.8 (minor), 132.7 (major), 129.9 (minor), 129.7 (major), 128.5 (major), 128.4 (minor), 53.8 (minor, N-CH), 51.3 (major, N-CH), 39.8 (minor, N-CH_2_), 39.5 (major, N-CH_2_), 33.5 (minor, CH_2_-C-CH_2_), 33.2 (major, CH_2_-C-CH_2_), 29.9 (minor, CH), 29.5 (major, CH), 27.0 (minor, C), 26.9 (major, C), 21.0 (major, CH_2_-C-CH_3_), 20.2 (major, CH_2_-C-CH_3_), 20.1 (minor, CH_3_), 19.9 (minor, CH_3_), 17.1 (major, CH_2_), 16.6 (minor, CH_2_), 13.83 (minor, CH_2_-CH_3_), 13.79 (major, CH_2_-CH_3_). HRMS (ESI) (*m/z*) calculated for C_16_H_20_ClNO [M + Na]^+^: 300.1126, found: 300.1127. IR *v*/cm^−1^ (KBr) 1676, 1497, 1455, 1417, 1376, 1091, 1013, 819, 723, 527.1-(4-Methoxyphenyl)-4-methyl-3-phenyl-3-azabicyclo[3.1.0]hexan-2-one (**2j**). White solid. (Yield: 72%). Mp = 48–50 ºC. dr ≈ 2:1. ^1^H NMR (400 MHz, CDCl_3_) δ 7.49 (d, *J* = 7.6 Hz, 2H, Ar-H), 7.42–7.32 (m, 6H, Ar-H), 7.25–7.13 (m, 2H, Ar-H), 6.93–6.84 (m, 3H, Ar-H), 4.59 (p, *J* = 6.0 Hz, 0.5H, minor, N-CH), 4.27 (q, *J* = 6.4 Hz, 1H, major, N-CH), 3.81 (s, 3H, major, Ar-OCH_3_), 3.80 (s, 1.4H, minor, Ar-OCH_3_), 2.34 (dt, *J* = 7.6, 4.8 Hz, 0.5H, minor, CH), 2.01 (dd, *J* = 7.6, 4.0 Hz, 1H, major, CH), 1.50 (dd, *J* = 7.6, 4.4 Hz, 1H, major, CH_2_), 1.41–1.36 (m, 4H), 1.29–1.24 (m, 1.5H, CH_2_), 1.23–1.19 (m, 2.8H). ^13^C NMR (151 MHz, CDCl_3_) δ 174.2 (minor, C=O), 173.3 (major, C=O), 158.9 (major), 158.7 (minor), 138.1 (major), 137.0 (minor), 130.2 (major), 129.9 (minor), 129.0 (major), 128.9 (minor), 128.4 (major), 128.3 (minor), 125.6 (minor), 125.1 (major), 124.5 (minor), 123.0 (minor), 114.0 (major), 113.8 (minor), 56.1 (major, Ar-OCH_3_), 55.4 (major, C-N), 55.3 (minor, Ar-OCH_3_), 53.0 (minor, C-N), 34.8 (major, C), 33.7 (minor, C), 26.6 (minor, CH), 26.2 (major, CH), 21.5 (major, CH_3_), 19.6 (major, CH_3_), 17.0 (minor, CH_2_), 16.2 (minor, CH_2_). HRMS (ESI) (*m/z*) calculated for C_19_H_19_NO_2_ [M + Na]^+^: 316.1308, found: 316.1308. IR *v*/cm^−1^ (KBr) 1682, 1516, 1492, 1392, 1369, 1296, 1177, 1107, 1031, 838, 761, 747, 538.1-(3-Methoxyphenyl)-4-methyl-3-phenyl-3-azabicyclo[3.1.0]hexan-2-one (**2k**). White solid. (Yield: 80%). Mp = 78–80 °C. dr ≈ 5:3. ^1^H NMR (600 MHz, CDCl_3_) δ 7.48 (d, *J* = 7.8 Hz, 2H, Ar-H), 7.37 (dd, *J* = 16.6, 8.9 Hz, 3H, Ar-H), 7.28 (t, *J* = 7.8 Hz, 1H, Ar-H), 7.26–7.15 (m, 4H, Ar-H), 7.11 (s, 1H, Ar-H), 7.04 (d, *J* = 7.8 Hz, 1H, major, Ar-H), 6.99 (d, *J* = 7.8 Hz, 0.5H, minor, Ar-H), 6.84 (dd, *J* = 8.4, 2.4 Hz, 1H, major, Ar-H), 6.81 (dd, *J* = 7.8, 2.1 Hz, 0.5H, minor, Ar-H), 4.59 (p, *J* = 6.0 Hz, 0.6H, minor, N-CH), 4.26 (q, *J* = 6.0 Hz, 1H, major, N-CH), 3.83 (s, 3H, major, Ar-OCH_3_), 3.81 (s, 1.7H, minor, Ar-OCH_3_), 2.40 (dt, *J* = 7.8, 5.0 Hz, 0.6H, minor, CH), 2.06 (dd, *J* = 7.8, 4.2 Hz, 1H, major, CH), 1.55 (dd, *J* = 7.8, 4.8 Hz, 1H, major, CH_2_), 1.44 (dd, *J* = 7.8, 4.8 Hz, 0.6H, minor, CH_2_), 1.39 (d, *J* = 6.6 Hz, 3H, major, CH_3_), 1.32 (t, *J* = 4.8 Hz, 0.6H, minor, CH_2_), 1.25 (t, *J* = 4.5 Hz, 1H, major, CH_2_), 1.21 (d, *J* = 6.0 Hz, 1.7H, minor, CH_3_). ^13^C NMR (101 MHz, CDCl_3_) δ 173.7 (minor, C=O), 172.8 (major, C=O), 159.64 (major), 159.59 (minor), 138.0 (major), 137.9 (major), 136.9 (minor), 129.4 (major), 129.3 (minor), 129.0 (major), 128.9 (minor), 125.8 (minor), 125.3 (major), 124.6 (minor), 123.2 (major), 120.9 (major), 120.3 (minor), 114.7 (major), 114.0 (minor), 113.0 (minor), 112.8 (major), 56.1 (minor, Ar-OCH_3_), 55.3 (major, C-N), 53.0 (minor, Ar-OCH_3_), 35.1 (major, C), 33.9 (minor, C), 27.1 (minor, CH), 26.5 (major, CH), 21.4 (major, CH_3_), 20.0 (major, CH_3_), 17.0 (minor, CH_2_), 16.9 (major, CH_2_). HRMS (ESI) (*m/z*) calculated for C_19_H_19_NO_2_ [M + Na]^+^: 316.1308, found: 316.1308. IR *v*/cm^−1^ (KBr) 1682, 1594, 1493, 1456, 1372, 1293, 1186, 1040, 1028, 939, 759, 623.4-Methyl-3-phenyl-1-(*p*-tolyl)-3-azabicyclo[3.1.0]hexan-2-one (**2l**). White solid. (Yield: 67%). Mp = 89–91 °C. dr ≈ 5:4. ^1^H NMR (600 MHz, CDCl_3_) δ 7.49 (d, *J* = 7.8 Hz, 2H, Ar-H), 7.39–7.34 (m, 7H, Ar-H), 7.25 (s, 0.7H, Ar-H), 7.20–7.13 (m, 5H, Ar-H), 4.59 (p, *J* = 6.0 Hz, 0.7H, minor, N-CH), 4.27 (q, *J* = 6.0 Hz, 1H, major, N-CH), 2.37–2.34 (m, 3.8H), 2.33 (s, 2H), 2.02 (dd, *J* = 7.8, 4.2 Hz, 1H, major, CH), 1.54 (dd, *J* = 7.8, 4.8 Hz, 1H, major, CH_2_), 1.42 (dd, *J* = 7.8, 4.8 Hz, 0.8H, minor, CH_2_), 1.39 (d, *J* = 6.0 Hz, 3H, major, CH_3_), 1.28 (t, *J* = 4.2 Hz, 1H, major, CH_2_), 1.23 (d, *J* = 4.2 Hz, 1H, major, CH_2_), 1.21 (d, *J* = 6.6 Hz, 2.4H, minor, CH_3_). ^13^C NMR (151 MHz, CDCl_3_) δ 174.1 (minor, C=O), 173.1 (major, C=O), 138.1 (major), 137.03 (major), 137.00 (minor), 136.7 (minor), 133.3 (major), 133.2 (minor), 129.2 (major), 129.1 (major), 128.95 (major), 128.85 (minor), 128.8 (minor), 128.5 (minor), 125.6 (minor), 125.1 (minor), 124.5 (major), 123.0 (major), 56.1 (major, C-N), 53.0 (minor), 35.0 (major, C), 33.9 (minor, C), 26.8 (minor, CH), 26.4 (major, CH), 21.5 (major, Ar-CH_3_), 21.15 (minor, Ar-CH_3_), 21.11 (minor, CH_3_), 19.6 (major, CH_3_), 17.0 (major, CH_2_), 16.2 (minor, CH_2_). HRMS (ESI) (*m/z*) calculated for C_19_H_19_NO [M + Na]^+^: 300.1359, found: 300.1358. IR *v*/cm^−1^ (KBr) 1678, 1595, 1493, 1456, 1371, 1296, 1179, 1110, 1066, 1031, 922, 762, 640, 529.4-Methyl-3-phenyl-1-(*m*-tolyl)-3-azabicyclo[3.1.0]hexan-2-one (**2m**). White solid. (Yield: 76%). Mp = 103–105 °C. dr ≈ 2:1. ^1^H NMR (400 MHz, CDCl_3_) δ 7.52–7.48 (m, 2H, Ar-H), 7.40–7.30 (m, 5H, Ar-H), 7.26–7.14 (m, 5H, Ar-H), 7.13–7.05 (m, 1.6H, Ar-H), 4.59 (p, *J* = 6.0 Hz, 0.6H, minor, N-CH), 4.27 (q, *J* = 6.4 Hz, 1H, major, N-CH), 2.39–2.35 (m, 3.8H), 2.34 (s, 1.7H, minor), 2.04 (dd, *J* = 8.0, 4.4 Hz, 1H, major, CH), 1.57–1.54 (m, 1H, CH_2_), 1.45 (dd, *J* = 7.6, 4.8 Hz, 0.6H, CH_2_), 1.40 (d, *J* = 9.6 Hz, 3H, major, CH_3_), 1.29 (t, *J* = 4.8 Hz, 0.6H, minor, CH_2_), 1.24 (d, *J* = 4.4 Hz, 1H, CH_2_), 1.22 (d, *J* = 6.0 Hz, 2H, minor, CH_3_). ^13^C NMR (101 MHz, CDCl_3_) δ 174.0 (C=O), 173.1 (C=O), 138.1, 137.9, 137.0, 136.2, 136.1, 129.7, 129.5, 129.0, 128.9, 128.4, 128.3, 128.1, 127.8, 125.9, 125.7, 125.4, 125.2, 124.5, 123.1, 56.1 (C-N), 53.0 (C-N), 35.2 (C), 34.1 (C), 26.9 (CH), 26.4 (CH), 21.5 (Ar-CH_3_), 21.4 (CH_3_), 19.6 (CH_3_), 17.0, (CH_2_), 16.2 (CH_2_). HRMS (ESI) (*m/z*) calculated for C_19_H_19_NO_2_ [M + Na]^+^: 300.1359, found: 300.1359. IR *v*/cm^−1^ (KBr) 1683, 1494, 1455, 1376, 1296, 1181, 1108, 1067, 758, 606, 511.4-Methyl-3-phenyl-1-(*o*-tolyl)-3-azabicyclo[3.1.0]hexan-2-one (**2n**). White solid. (Yield: 62%). Mp = 100–102 °C. dr ≈ 5:3. ^1^H NMR (600 MHz, CDCl_3_) δ 7.47 (d, *J* = 7.8 Hz, 2H, Ar-H), 7.40–7.31 (m, 4H, Ar-H), 7.27 (s, 0.5H, Ar-H), 7.25–7.14 (m, 7H, Ar-H), 4.66 (p, *J* = 6.0 Hz, 0.6H, minor, N-CH), 4.33 (q, *J* = 6.0 Hz, 1H, major, N-CH), 2.54 (s, 3H, major, Ar-CH_3_), 2.40 (s, 1.7H, minor, Ar-CH_3_), 2.22 (dt, *J* = 7.8, 4.8 Hz, 0.6H, minor, CH), 2.06 (dd, *J* = 7.2, 3.6 Hz, 1H, major, CH), 1.52 (dd, *J* = 7.8, 4.8 Hz, 1H, major, CH_2_), 1.47 (d, *J* = 4.8 Hz, 0.5H, minor, CH_2_), 1.45 (d, *J* = 6.0 Hz, 3H, major, CH_3_), 1.33–1.29 (m, 2H, CH_2_), 1.25 (d, *J* = 6.6 Hz, 2.4H, minor, CH_3_). ^13^C NMR (151 MHz, CDCl_3_) δ 173.9 (minor, C=O), 172.8 (major, C=O), 139.8 (major), 139.2 (minor), 138.0 (major), 137.1 (minor), 134.6 (minor), 134.3 (major), 130.7 (minor), 130.6 (major), 130.5 (major), 130.2 (minor), 128.95 (major), 128.89 (minor), 128.0 (major), 127.9 (minor), 125.9 (minor), 125.74 (major), 125.70 (minor), 125.2 (major), 124.5 (minor), 123.1 (major), 56.3 (major, C-N), 53.2 (minor, C-N), 35.5 (major, C), 34.7 (minor, C), 26.2 (minor, CH), 26.0 (major, CH), 20.9 (major, Ar-CH_3_), 20.1 (minor, Ar-CH_3_), 19.6 (minor, CH_3_), 19.5 (major, CH_3_), 17.1 (major, CH_2_), 14.6 (minor, CH_2_). HRMS (ESI) (*m/z*) calculated for C_19_H_19_NO_2_ [M + Na]^+^: 300.1359, found: 300.1358. IR *v*/cm^−1^ (KBr) 1682, 1596, 1495, 1456, 1373, 1293, 1179, 1099, 1067, 1027, 923, 751, 728, 659, 533.4-Methyl-1,3-diphenyl-3-azabicyclo[3.1.0]hexan-2-one (**2o**). Yellow solid. (Yield: 83%). Mp = 87–89 °C. dr ≈ 3:2. ^1^H NMR (400 MHz, CDCl_3_) δ 7.52–7.46 (m, 5H, Ar-H), 7.41–7.33 (m, 6H, Ar-H), 7.32–7.28 (m, 1.2H, Ar-H), 7.28–7.26 (m, 1H, Ar-H),7.25–7.23 (m, 0.7H, Ar-H), 7.22–7.15 (m, 1.6H, Ar-H, 4.60 (p, *J* = 6.0 Hz, 0.7H, minor, N-CH), 4.28 (q, *J* = 6.3 Hz, 1H, major, N-CH), 2.43–2.37 (m, 0.7H, minor, CH), 2.07 (dd, *J* = 7.6, 4.4 Hz, 1H, major, CH), 1.58–1.55 (m, 1H, major, CH_2_), 1.46 (dd, *J* = 7.6, 4.8 Hz, 0.7H, minor, CH_2_), 1.39 (d, *J* = 6.0 Hz, 3H, major, CH_3_), 1.32 (t, *J* = 4.8 Hz, 0.7H, minor, CH_2_), 1.26 (t, *J* = 4.8 Hz, 1H, major, CH_2_), 1.22 (d, *J* = 6.0 Hz, 2H, minor, CH_3_). ^13^C NMR (151 MHz, CDCl_3_) δ 173.9 (minor, C=O), 173.0 (major, C=O), 138.0 (major), 136.9 (major), 136.3 (minor), 136.2 (minor), 129.0 (minor), 128.9 (major), 128.50, 128.47, 128.4, 127.3, 127.0, 125.7, 125.2, 124.6, 123.1, 56.1 (major, C-N), 53.0 (minor, C-N), 35.2 (major, C), 34.1 (minor, C), 26.9 (minor, C-H), 26.4 (major, C-H), 21.5 (major, CH_2_), 19.7 (minor, CH_2_), 17.0 (major, CH_3_), 16.4 (minor, CH_3_). HRMS (ESI) (*m/z*) calculated for C_18_H_17_NO [M + Na]^+^: 286.1202, found: 286.1202. IR *v*/cm^−1^ (KBr) 1682, 1598, 1495, 1447, 1372, 1299, 1178, 1101, 1063, 1021, 757, 664, 530.1-(4-Fluorophenyl)-4-methyl-3-phenyl-3-azabicyclo[3.1.0]hexan-2-one (**2p**). White solid. (Yield: 79%). Mp = 98–100 °C. dr ≈ 2:1. ^1^H NMR (600 MHz, CDCl_3_) δ 7.48 (d, *J* = 7.8 Hz, 1.6H, Ar-H), 7.47–7.43 (m, 2.4H, Ar-H), 7.37 (q, *J* = 7.8 Hz, 2.5H, Ar-H), 7.24 (d, *J* = 7.8 Hz, 1H, Ar-H), 7.22–7.16 (m, 1.2H, Ar-H), 7.05 (t, *J* = 8.7 Hz, 1.6H, major, Ar-H), 7.02 (t, *J* = 8.7 Hz, 1H, minor, Ar-H), 4.59 (p, *J* = 6.0 Hz, 0.4H, minor, N-CH), 4.27 (q, *J* = 6.0 Hz, 1H, major, N-CH), 2.38 (dt, *J* = 7.8, 5.1 Hz, 0.4H, minor, CH), 2.05 (dd, *J* = 7.8, 4.2 Hz, 1H, major, N-CH), 1.51 (dd, *J* = 7.8, 4.2 Hz, 1H, major, CH_2_), 1.39 (d, *J* = 6.0 Hz, 3H, major, CH_3_), 1.31 (t, *J* = 4.5 Hz, 0.5H, minor, CH_2_), 1.26 (t, *J* = 4.5 Hz, 1H, major, CH_2_), 1.22 (d, *J* = 6.0 Hz, 1.3H, minor, CH_3_). ^13^C NMR (101 MHz, CDCl_3_) δ 173.8 (minor, C=O), 172.8 (major, C=O), 162.1 (d, *J* = 246.4 Hz, major, C-F), 162.0 (d, *J* = 246.4 Hz, minor, C-F), 137.9 (major), 136.8 (minor), 132.1 (d, *J* = 3.0 Hz, major), 132.0 (d, *J* = 4.0 Hz, minor), 130.6 (d, *J* = 8.1 Hz, major), 130.3 (d, *J* = 8.1 Hz, minor), 129.0 (major), 128.9 (minor), 125.8 (minor), 125.3 (major), 124.5 (minor), 123.2 (major), 115.3 (d, *J* = 22.2 Hz, major), 115.2 (d, *J* = 21.2 Hz, minor), 56.1 (major, C-N), 53.0 (minor, C-N), 34.7 (major, C), 33.6 (minor, C), 26.7 (minor, C-H), 26.3 (major, C-H), 21.5 (major, CH_2_), 19.9 (minor, CH_2_), 16.9 (major, CH_3_), 16.5 (minor, CH_3_). ^19^F NMR (376 MHz, CDCl_3_) δ -115.1 (major), -115.5 (minor). HRMS (ESI) (*m/z*) calculated for C_18_H_16_FNO [M + Na]^+^: 304.1108, found: 304.1107. IR *v*/cm^−1^ (KBr) 1685, 1493, 1455, 1406, 1394, 1381, 1250, 1230, 1076, 1066, 1028, 757, 600.1-(4-Bromophenyl)-4-methyl-3-phenyl-3-azabicyclo[3.1.0]hexan-2-one (**2q**). White solid. (Yield: 80%). Mp = 119–121 °C. dr ≈ 5:3. ^1^H NMR (600 MHz, CDCl_3_) δ 7.49 (d, *J* = 8.4 Hz, 2H, Ar-H), 7.46 (t, *J* = 8.1 Hz, 3H, Ar-H), 7.40–7.34 (m, 6.5H, Ar-H), 7.24 (d, *J* = 7.8 Hz, 1H, Ar-H), 7.19 (dt, *J* = 11.4, 7.5 Hz, 1.6H, Ar-H), 4.58 (p, *J* = 6.0 Hz, 0.6H, minor, N-CH), 4.27 (q, *J* = 6.0 Hz, 1H, major, N-CH), 2.39 (dt, *J* = 7.8, 5.1 Hz, 0.6H, minor, CH), 2.06 (dd, *J* = 7.2, 4.8 Hz, 1H, major, CH), 1.52 (dd, *J* = 7.2, 4.8 Hz, 1H, major, CH_2_), 1.41 (dd, *J* = 7.8, 4.8 Hz, 1H, major, CH_2_), 1.38 (d, *J* = 6.6 Hz, 3H, major, CH_3_), 1.33 (t, *J* = 4.8 Hz, 0.7H, minor, CH_2_), 1.28 (t, *J* = 4.8 Hz, 1H, major, CH_2_), 1.21 (d, *J* = 6.6 Hz, 1.8H, minor, CH_3_). ^13^C NMR (101 MHz, CDCl_3_) δ 173.4 (minor, C=O), 172.4 (major, C=O), 137.8 (major), 136.7 (minor), 135.42 (major), 135.36 (minor), 131.6 (major), 131.4 (minor), 130.5 (major), 130.1 (minor), 129.0 (major), 128.9 (minor), 125.9 (minor), 125.4 (major), 124.6 (minor), 123.2 (major), 121.3 (major), 121.0 (minor), 56.1 (major, C-N), 53.0 (minor, C-N), 34.6 (major, C), 33.5 (minor, C), 26.9 (minor, CH), 26.5 (major, CH), 21.5 (major, CH_3_), 20.1 (minor, CH_3_), 16.9 (major, CH_2_), 16.8 (minor, CH_2_). HRMS (ESI) (*m/z*) calculated for C_18_H_16_BrNO [M + Na]^+^: 364.0307, found: 364.0307. IR *v*/cm^−1^ (KBr) 1682, 1489, 1389, 1382, 1100, 1066, 1057, 765, 699.4-Benzyl-1-(4-chlorophenyl)-3-(*p*-tolyl)-3-azabicyclo[3.1.0]hexan-2-one: (**2r**). White solid. (Yield: 81%). Mp = 92–93 °C. dr ≈ 10:1. ^1^H NMR (600 MHz, CDCl_3_) δ 7.50 (d, *J* = 7.8 Hz, 2H, Ar-H), 7.43 (d, *J* = 7.8 Hz, 0.2H, Ar-H), 7.37 (d, *J* = 7.8 Hz, 0.5H, Ar-H), 7.36–7.27 (m, 3.5H, Ar-H), 7.24 (d, *J* = 7.8 Hz, 2.4H, Ar-H), 7.23–7.16 (m, 2.4H, Ar-H), 7.16–7.10 (m, 1.8H, Ar-H), 4.60–4.55 (m, 0.2H, N-CH), 4.49 (dd, *J* = 6.0, 3.4 Hz, 1H, N-CH), 3.01 (d, *J* = 13.8, 3.6 Hz, 1H, CH_2_), 2.94 (dd, *J* = 13.8, 6.6 Hz, 1H, CH_2_), 2.48–2.41 (m, 0.3H, CH_2_), 2.37 (s, 3H, major, CH_3_), 2.34 (s, 0.3H, minor, CH_3_), 2.31–2.45 (m, 0.2H, CH_2_), 2.17–2.12 (m, 0.2H, CH_2_), 2.06 (dd, *J* = 7.8, 4.8 Hz, 1H, CH_2_), 1.62–1.56 (m, 0.3H, CH_2_),1.52 (dd, *J* = 7.8, 5.4 Hz, 0.1H, CH_2_), 1.46 (dd, *J* = 7.7, 4.9 Hz, 1H), 1.27 (t, *J* = 4.4 Hz, 1H), 1.22–1.15 (m, 1H). ^13^C NMR (151 MHz, CDCl_3_) δ 173.5 (minor, C=O), 172.8 (major, C=O), 137.2, 136.1, 135.9, 135.2, 135.1, 134., 134.50, 134.1, 133.0, 132.9, 130.2, 129.9, 129.81, 129.78, 129.7, 129.1, 128.9, 128.7, 128.4, 128.3, 126.95, 126.88, 124.8, 122.9, 60.6 (major, N-CH), 59.2 (minor, N-CH), 39.2 (major, Ar-CH_2_), 37.5 (minor, Ar-CH_2_), 34.9 (major, CH), 33.8 (minor, CH), 24.9 (minor, C), 24.0 (major, C), 21.1 (minor, CH_3_), 21.0 (major, CH_3_), 19.0 (major, CH_2_), 17.3 (minor, CH_2_). HRMS (ESI) (*m/z*) calculated for C_25_H_22_ClNO [M + Na]^+^: 410.1282, found: 410.1275. IR *v*/cm^−1^ (KBr) 1681, 1514, 1494, 1384, 1293, 1082, 1066, 836, 749, 726, 531.

## 4. Conclusions

In summary, we developed a base-promoted intramolecular alkene addition reaction starting from readily available vinyl cyclopropanes to access a series of conformationally restricted biologically valuable highly substituted aza[3.1.0]bicycles in moderate to good yields. The transformation was performed in the presence of *^t^*BuOK in DMF at 110 °C under an air atmosphere. Experiments showed that large concentrations of the base are beneficial to the nucleophilic addition process. Although the protocol is limited to substituted cyclopropionamides with a range of functional aryl groups, the cyclopropane moiety in the fused ring is a valuable derivatization unit for the further construction of structurally diverse biologically organic molecules. This reaction was tailor-made for saturated aza[3.1.0]bicycle-containing fused bicyclic compounds. Further derivatization and chemical biology application evaluation of aza[3.1.0]bicyclic compounds are concurrently underway in our laboratory.

## Data Availability

Not applicable.

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
