# Peer review of "Base-Promoted Intramolecular Addition of Vinyl Cyclopropanecarboxamides to Access Conformationally Restricted Aza[3.1.0]bicycles"

_molecules, 2023, doi:10.3390/molecules28093691_

Round 1

Reviewer 1 Report

The authors present tBuOK-promoted intramolecular addition of vinyl cyclopropanecarboxamides to synthesize conformationally restricted aza[3.1.0]bicycles core. A wide range of biologically valuable highly substituted aza[3.1.0]bicycles could ben obtained in moderate to good yields through the direct C-N bond formation.  The protocol features simple operation, metal and external oxidant-free conditions. Considering the operational simplicity of the procedure, this protocol should be attractive to the organic synthesis community. I recommend publication of the work in Molecules after addressing some minor points:

1. Some white-space, bold and typographical errors need to be further revised.

2. What about the yield of 2a if the loading of tBuOK was decreased to 1equiv or 2 equiv.

3. The note b in schemes 2 and 3 should be italic.

4. The structure of products in the manuscript should be consistent with supporting information. 

Author Response

  1. Some white-space, bold and typographical errors need to be further revised.

Reply: We have revised them accordingly.

  1. What about the yield of 2a if the loading of tBuOK was decreased to 1equiv or 2 equiv.

Reply: the yield of 2a was isolated in 31% (63%)、64% (35%) and 72% (6%), when we performed the reaction in the presence of 1.0 equiv., 2.0 equiv., and 3.0 equiv. of tBuOK.

  1. The note b in schemes 2 and 3 should be italic.

Reply: We have revised them accordingly.

  1. The structure of products in the manuscript should be consistent with supporting information.

Reply: We have revised them accordingly.

Reviewer 2 Report

Guisheng Zhang and coworkers described intersting protocol for the preparation of wide range of Aza[3.1.0]bicyclic molecular segments. This topic is interesting and the manuscript is written rather well. Some weak points of this contribution must be however improved before the publication. In particular:

+ "T" is a symbol dedicated for temperatures in Kelvin scale. For Celsious scale, th symbol "t" is dediceted.

+ The column 4 within the table 1 have not sens.

+ In my opinion, Scheme 4 have not sens.

+ Paragraph 3.1:
Mobile phase for TLC experiments should be defined.

+ Paragraph 3.3:
(a) For full characterisation of synthetised compounds, their IR spectra should be presented. In particular, key signals should be assigned for respective functional groups and molecular segments.
(b) Within NMR spectrums, key signals should be assigned for respective atoms in molecules.
(c) Do obtained solid products were recrystallised before the measuring of m.p.'s?

Author Response

+ "T" is a symbol dedicated for temperatures in Kelvin scale. For Celsious scale, th symbol "t" is dedicated

Reply: We have revised them accordingly.

+ The column 4 within the table 1 have not sens.

Reply: We have deleted it.

+ In my opinion, Scheme 4 have not sens.

Reply: Gram-scale preparation may be useful for certain researchers or in some fields. So we're not going to delete it.

+ Paragraph 3.1:

Mobile phase for TLC experiments should be defined.

Reply: Mobile phase for TLC experiments have been defined.

+ Paragraph 3.3:

(a) For full characterisation of synthetised compounds, their IR spectra should be presented. In particular, key signals should be assigned for respective functional groups and molecular segments.

Reply: IR spectra have been added in the “3. Materials and Methods”. Please see the revised manuscript for more information.

(b) Within NMR spectrums, key signals should be assigned for respective atoms in molecules.

Reply: The key signals have been assigned. Please see the revised manuscript for more information.

(c) Do obtained solid products were recrystallised before the measuring of m.p.'s?

Reply: No. They were purified by flash column chromatography.

Reviewer 3 Report

With reference to manuscript ID molecules-2357704 titled " Base-Promoted Intramolecular Addition of Vinyl Cyclopropanecarboxamides to Access Conformationally Restricted Aza[3.1.0]bicycles". Author did a good peace of work.  The introduction is concise and well addressed. Research methodology is appropriate and justified. The synthesized compounds have been well characterized by spectral techniques. The manuscript is quite informative. I suggest to accept the manuscript for the publication. 

Author Response

Reply: Thank you for your comments and suggestions.